# Effects of Zinc on the Right Cardiovascular Circuit in Long-Term Hypobaric Hypoxia in Wistar Rats

**DOI:** 10.3390/ijms24119567

**Published:** 2023-05-31

**Authors:** Karem Arriaza, Julio Brito, Patricia Siques, Karen Flores, Stefany Ordenes, Daniel Aguayo, María del Rosario López, Silvia M. Arribas

**Affiliations:** 1Institute of Health Studies, University Arturo Prat, Av. Arturo Prat 2120, Iquique 1110939, Chile; jbritor@tie.cl (J.B.); psiques@tie.cl (P.S.); kfloresu@unap.cl (K.F.); stefany.ordenes@gmail.com (S.O.); daguayo180@gmail.com (D.A.); 2Institute DECIPHER, German-Chilean Institute for Research on Pulmonary Hypoxia and Its Health Sequelae, Hamburg (Germany) and Iquique (Chile), Avenida Arturo Prat 2120, Iquique 1110939, Chile; 3Department of Physiology, Faculty of Medicine, University Autónoma of Madrid, 28029 Madrid, Spain; mrosario.lopez@uam.es (M.d.R.L.); silvia.arribas@uam.es (S.M.A.)

**Keywords:** right ventricle hypertrophy, hypobaric hypoxia, zinc, hypoxic pulmonary vasoconstriction

## Abstract

Hypobaric hypoxia under chromic conditions triggers hypoxic pulmonary vasoconstriction (HPV) and right ventricular hypertrophy (RVH). The role of zinc (Zn) under hypoxia is controversial and remains unclear. We evaluated the effect of Zn supplementation in prolonged hypobaric hypoxia on HIF2α/MTF-1/MT/ZIP12/PKCε pathway in the lung and RVH. Wistar rats were exposed to hypobaric hypoxia for 30 days and randomly allocated into three groups: chronic hypoxia (CH); intermittent hypoxia (2 days hypoxia/2 days normoxia; CIH); and normoxia (sea level control; NX). Each group was subdivided (n = 8) to receive either 1% Zn sulfate solution (z) or saline (s) intraperitoneally. Body weight, hemoglobin, and RVH were measured. Zn levels were evaluated in plasma and lung tissue. Additionally, the lipid peroxidation levels, HIF2α/MTF-1/MT/ZIP12/PKCε protein expression and pulmonary artery remodeling were measured in the lung. The CIH and CH groups showed decreased plasma Zn and body weight and increased hemoglobin, RVH, and vascular remodeling; the CH group also showed increased lipid peroxidation. Zn administration under hypobaric hypoxia upregulated the HIF2α/MTF-1/MT/ZIP12/PKCε pathway and increased RVH in the intermittent zinc group. Under intermittent hypobaric hypoxia, Zn dysregulation could participate in RVH development through alterations in the pulmonary HIF2α/MTF1/MT/ZIP12/PKCε pathway.

## 1. Introduction

Hypobaric hypoxia occurs when there is a decrease in the supply of oxygen to tissues due to a drop in oxygen partial pressure caused by exposure to a low-pressure atmosphere, such as at high altitudes [1]. Exposure to hypobaric hypoxia triggers a series of adaptive changes, with the most relevant modifications occurring in the right cardiopulmonary system [2,3]. These changes may also be maladaptive when hypoxic stimulation is persistent and are associated with several pathological conditions, such as high-altitude pulmonary hypertension (HAPH) and subsequent right ventricular hypertrophy (RVH) [4]. This problem is relevant, considering that it has been estimated that approximately 81.6 million people currently live and work at high altitudes of over 2500 m above sea level (m.a.s.l.) [5].

There are different classifications of hypobaric hypoxia depending on the exposure time at a high altitude. Acute hypobaric hypoxia exposure occurs after a short duration of exposure (hours or days) and primarily involves tourists and mountaineers. Chronic hypobaric hypoxia (CH) is characterized by permanent exposure to high altitudes, such as populations living at high altitudes (Andeans, Tibetans, and Sherpas, among others) [6,7]. Some years ago, long-term chronic intermittent hypoxia (CIH) was described as a new condition of hypobaric hypoxia exposure, where individuals commute to work at a high altitude for a few days and then return to sea level for a period of rest [6]. Subjects exposed to CIH conditions may experience an exacerbated response to hypoxic pulmonary vasoconstriction (HPV), one of the first physiological compensatory mechanisms in pulmonary circulation, which allows the blood flow to redistribute to more ventilated areas of the lung for better gas exchange [8,9]. However, in the long term, prolonged vasoconstriction induces the structural remodeling of pulmonary arterioles, subsequently leading to the chronic elevation of pulmonary artery pressure and the development of HAPH and RVH [4,10,11].

The beneficial effects of Zn supplementation on inflammation and oxidative stress are still inconclusive. On one hand, Zn is an important co-factor and a structural component of antioxidant enzymes such as SOD [12]. It can also inhibit ROS-producing enzymes such as NADPH-oxidase and reduce chronic inflammation [13,14]. Zn supplementation has been successfully used in some situations with Zn deficiency [15], and recent meta-analysis has supported the beneficial-anti-inflammatory and antioxidative effects in adults [16]. However, there is controversy over the effects of Zn supplements on cardiovascular risk factors and events since Zn can also have prooxidant actions, as a transition metal [17]. Among others, it can act in redox-dependent signaling cascades in which MT is implicated, which has been suggested as a “link between redox and zinc signaling” [18,19]. It is also important to note that zinc can induce MAPKs, and we have previously demonstrated that the elevation of p38 MAPK expression and activity is associated with RVH [20]. We have incorporated this controversy in the introduction and discussion sections.

In 2008, Bernal et al. [21] uncovered a new HPV pathway that is related to alterations in Zn metabolism. The authors demonstrated that metallothionein (MT) plays a key role in this pathway. MT is a metal-binding, zinc storage protein, but this protein releases Zn through posttranslational regulation under hypoxic conditions. This release of Zn activates PKCε, which has been implicated in contractile events [22,23] and contributes to HPV [21]. These data are supported by a study showing that PKCε-deficient animals exhibit reduced HPV [24]. Additionally, PKCε activation has been associated with elevated cytoplasmic Zn levels under hypobaric hypoxic conditions. This process is also related to oxidative stress induced by an excess of reactive oxygen species (ROS), the levels of which are not compensated for by the antioxidant system [25]. In fact, oxidative stress and inflammation are relevant mechanisms in the development of hypoxia-associated pathologies, including pulmonary artery remodeling and RVH [26]. Under hypoxic conditions, ROS are released and accumulate, as demonstrated in cells and tissues. For example, an increase in the cerebral output of ROS under hypoxic conditions has been proposed to have implications for acute mountain sickness [27]. Alterations in oxygen consumption by leucocytes under hypoxia have been demonstrated in vitro and at high altitude [28] and increased oxidative damage has been found after both acute and chronic high-altitude exposure [29].

In the molecular pathways previously described, other authors have established that MT activation under hypoxic conditions (normobaric hypoxia) requires the overexpression of transcription factors, such as hypoxia-inducible factor 1 alpha (HIF1α) and metal-responsive transcription factor (MTF-1) [30,31,32]. Moreover, another study on chronic hypoxia showed that the overexpression of both HIF1α and HIF2α promoted transcription of the Zn transporter (namely, zinc-regulated, iron-regulated transporter-like protein; ZIP12) through hypoxia regulatory elements (HREs). In addition, the overexpression of ZIP12 has been associated with the development of pulmonary hypertension and RVH, both of which are reduced by ZIP12 suppression [11].

Thus, the role of Zn in the alterations of the cardiopulmonary circuit under hypoxic conditions is controversial. Moreover, CIH is a relevant condition in which the role of Zn has not been explored. Therefore, the present study was designed with the aim of evaluating the implication of the Zn pathway HIF2α/MTF-1/MT/ZIP12/PKCε on RVH/cardiopulmonary system alterations induced by chronic and intermittent hypobaric hypoxia.

## 2. Results

### 2.1. Body Weight and Hemoglobin

The NX groups exhibited an increase in body weight throughout the 30-day period (*p* < 0.05), and there were no differences between the saline- and Zn-treated groups. However, the rats exposed to hypobaric hypoxia (CIH and CH) showed a reduction in body weight from day 0 to day 30 of exposure in both groups administered saline and Zn (*p* < 0.05), without any effect from Zn treatment. With respect to the hemoglobin concentration, the results showed an increase in all the groups exposed to hypobaric hypoxia at the end of the exposure (day 30) compared to the NX control groups in both the saline- and Zn-treated groups (*p* < 0.05), without an effect from Zn. The CH groups showed a greater increase in hemoglobin content than the CIH groups (*p* < 0.05) (Table 1).

### 2.2. Plasma and Lung Zinc Levels

The plasma Zn level was significantly decreased at day 30 in all groups exposed to hypobaric hypoxia (CIH and CH) compared to day 0 (*p* < 0.05). At day 30, plasma Zn levels were significantly lower in all exposed groups (CIH and CH) compared to the NX-matched controls (*p* < 0.05), with no differences between CIH and CH (*p* > 0.05). In NX rats, Zn administration did not influence Zn levels. However, the animals in the (CH)z group had significantly higher levels of plasma Zn than those in the (CH)s group of rats (*p* < 0.05), which was even lower at the beginning. The Zn level in (CIH)z-treated rats tended to be higher but was not significantly different compared to that of (CIH)s-treated rats (Table 1).

At day 30, the lung Zn levels were significantly lower in the exposed groups administered Zn (CIH and (CH)z) than in the (NX)z group (*p* < 0.05) (Table 1).

### 2.3. Right Ventricular Hypertrophy

RVH was observed in the rats under hypobaric hypoxia conditions (CIH and CH), both with saline and Zn administration, at the end of the exposure period (day 30). The CH group showed a higher degree of hypertrophy than the CIH group (*p* < 0.05). Zn administration did not have an effect on the NX group. There was a significantly larger Fulton’s index and myocyte area in the (CIH)z group than in the (CIH)s group (*p* < 0.05), but no significant differences were detected between the (CH)z and (CH)s groups (*p* > 0.05) (Figure 1A,B).

### 2.4. HIF2α, MTF-1, MT, ZIP12, and PKCε Protein Expression Levels

In lung tissue, HIF2α protein expression was not significantly different in any of the exposed groups (CIH and CH) treated with saline compared to the NX group (*p* > 0.05). Zn administration did not influence HIF2α levels in NX rats (*p* > 0.05). However, the levels were significantly greater in the (CIH)z group than in the (CIH)s and NX groups (*p* < 0.05). HIF2α levels tended to increase in (CH)z compared to (CH)s but did not reach statistical significance (*p* > 0.05) (Figure 2A).

MTF-1 expression was significantly higher in all the hypobaric hypoxic groups (CIH and CH) treated with saline and zinc compared with the control NX (*p* < 0.05), but no effects from Zn were observed. There were no differences between the CIH and CH groups (*p* > 0.05) (Figure 2B).

Hypoxia did not modify MT expression in the CIH and CH groups with saline administration compared to the NX group (*p* > 0.05). Zn administration did not modify MT expression in NX (*p* > 0.05). However, expression was higher in the (CH)z group than in the (CH)s group (*p* < 0.05). The MT level tended to be higher in (CIH)z than in (CIH)s, but the difference was not statistically significant (Figure 2C).

Hypoxia significantly increased the ZIP12 levels in the CH group after both saline and Zn administration compared to their NX groups (*p* < 0.05), and there were no differences between the CIH and CH groups. There was no effect from Zn administration. In the CIH group, ZIP12 levels also tended to be higher, although the difference was not significant (*p* > 0.05) (Figure 3A).

Hypoxia significantly increased PKCε levels in the (CH)s group compared to the (NX)s group (*p* < 0.05); levels also tended to be higher in (CIH)s-treated rats, but the difference was not significant (*p* > 0.05). The hypoxic groups treated with Zn showed overexpression of PKCε compared to (NX)z (*p* < 0.05); additionally, these levels in the (CIH)z group tended to be higher (Figure 3B).

### 2.5. Pulmonary Artery Remodeling

Hypobaric hypoxia significantly increased the total number of PASMC in both CIHs and CHs groups compared to the (NX)s group (*p* < 0.05). The hypoxic groups treated with Zn, only in the (CIH)z group, showed a significant increase in the PASMCs, compared to (NX)z (*p* < 0.05). In the (CH)z group, it tended to be higher without significance (Figure 4A,B).

Hypobaric hypoxia (CIH and CH) significantly increased the pulmonary artery wall thickness, without changes by zinc administration compared to NX groups (*p* < 0.05) (Figure 5A,B).

### 2.6. Lipid Peroxidation

The concentration of lipid peroxides, expressed as the level of MDA, was significantly increased only in the CH groups compared to the NX and CIH groups (*p* < 0.05). Zn administration did not have an additional effect on MDA levels (Figure 6).

## 3. Discussion

The present work evaluated the effects of Zn administration on cardiopulmonary alterations in rats in the context of hypobaric chronic and intermittent hypoxia with the following key findings. (1) CH exhibited larger degree of cardiopulmonary alterations than CIH, which were not substantially modified by Zn supplementation; (2) in CIH rats supplemented with Zn, there was a higher degree of RVH and increased PASMC, together with an overexpression of HIF2α, MTF-1, MT, and PKCε, which highlighted that Zn exposure increased cardiopulmonary damage with similar effects to those observed after chronic hypobaric hypoxia exposure under intermittent hypobaric hypoxia; and (3) ZIP12 overexpression seemed to be due to hypoxia and not to the influence of Zn.

Regarding general variables, we found a loss of body weight in all the hypobaric hypoxia groups, as previously reported in other studies [33,34]. In our study, Zn supplementation did not have a noticeable influence on body weight under hypoxic conditions. Importantly, both normoxic groups (treated with saline solution or zinc) ate the same amount of food (15 g per day), which was the total amount offered. However, the groups exposed to hypobaric hypoxia (CIH and CH) always left some uneaten food, averaging an intake of 10 g per day. Therefore, the weight loss in the groups exposed to hypoxia could be attributed to their lower food intake. In addition, at the end of the exposure period, there were no differences in weight between the saline and zinc groups, suggesting that Zn supplementation did not affect body weight, and this effect was instead due to hypoxia exposure. 

Another variable that was subject to changes due to exposure to hypobaric hypoxia was hemoglobin, which, in this study, significantly increased under hypobaric hypoxic conditions, as previously reported in other studies [35,36,37,38], without an effect from Zn administration. We did not find a greater increase in hemoglobin levels due to the exogenous administration of Zn, even though some studies have shown that zinc administration stimulates erythropoietin (EPO) under normoxic conditions in different study models [39,40]. However, Baranauskas et al. [41] reported that Zn administration under normobaric hypoxia (12 h of hypoxia) did not influence EPO levels. Therefore, this could explain why Zn in hypobaric hypoxic conditions did not contribute to the increase in hematological parameters.

We observed that Zn administration did not modify plasma levels of Zn in NX rats, suggesting that Zn levels were properly regulated under physiological conditions. However, under hypobaric hypoxia, we found a decrease in plasma Zn levels, regardless of exogenous Zn administration at a dose substantially lower than the reported LD50 [42,43]. Our findings were consistent with previous reports in humans exposed to acute high altitude [44] and in rats exposed to intermittent hypobaric hypoxia [45]. On the other hand, some controversial studies on Zn supplementation in acute hypoxia found an increase in serum Zn levels [41]; these data were in contrast to the results from our study, suggesting that differences in plasma zinc levels were more likely related to the time of exposure to hypoxia. We also found lower Zn levels in lungs from the CIH and CH groups supplemented with Zn. We suggest that the decreases in the Zn levels found in the plasma and lung tissues in the hypoxic groups in our research could be due to higher Zn urinary excretion under stress conditions or after exposure to high altitude, due to muscle breakdown [46]. The muscle contains the largest reserve of Zn in the body at approximately 57% [47], and, in cases of extreme necessity, there is mobilization from areas of lower requirement, such as bone, toward the skeletal muscles [48,49]. In addition, plasma levels are highly affected by factors such as circadian fluctuations and cytokine-related influences [50,51,52]. Therefore, under hypoxia, Zn administration apparently produces an alteration in Zn metabolism and mobilization.

RVH was found in all hypoxic groups with and without zinc, being higher in CH, as observed by hemoglobin levels, which was consistent with previously reported data in chronic and intermittent hypoxia [53,54]. Cardiac hypertrophy is characterized by an increase in the size of cardiomyocytes and thickening of the walls [55]. Interestingly, we observed a higher degree of enlarged cardiomyocytes in the (CIH)z group than in the (CIH)s group, suggesting that the supplementation promoted the hypertrophic effect in this group. Our data indicated that the exogenous administration of Zn could contribute to the development of RVH to a similar extent as that observed under prolonged hypoxic conditions under intermittent hypobaric hypoxia, eventually causing damage to right heart function and even leading to right heart failure [56]. Our data were also supported by the study by Zhao et al. [11], who found that the Zn pathways in prolonged hypoxia contribute to the development of pulmonary hypertension and RVH. It is important to mention that the data in our study contributed to novel findings in intermittent hypoxic conditions. With respect to chronic hypoxia conditions, these showed a higher degree of RVH and increased cardiomyocyte area, but without the effects of zinc administration. We think that the group chronically exposed to hypoxia already had a higher level of basal damage than the intermittent group and was not further damaged by the Zn supplementation. Although, we also found alterations in the Zn pathway responses in the CH (ZIP12 protein and PKCε activity). They converged to the same point and increased ROS, which were already elevated by hypoxia itself, as we have previously demonstrated in pulmonary arteries showing higher ROS production in CH than in CIH, and eNOS expression with NO availability were reduced only in CH [57]. This would contribute to a higher level of HPV and thus RVH. Therefore, we suggest that hypoxia was the main factor contributing to the damage under CH conditions. This was different to CIH, which had a worse response to hypoxia in the presence of Zn, likely due to the fact that this was a milder situation of hypoxia since CIH conditions involve intermittent and acute episodes of hypoxia, which result in a turn-on–turn-off regime for biological responses [58]. Therefore, zinc regulation could be time dependent on hypoxia and exert detrimental effects. 

At the molecular level, our research showed that HIF2α increased in the exposed groups (CIH and CH) only with the Zn administration but not under hypoxia alone. These data indicated the importance of the action of zinc added to the hypoxia factor on the stabilization of HIF2α at the pulmonary level. Previous studies have documented that HIF2α protein levels tended to stabilize and be overexpressed with a longer exposure time to hypoxia, which was why it was considered to play a significant role in cardiovascular pathologies in chronic hypoxia [59,60,61,62,63,64,65,66]. Supporting these findings, it had been demonstrated that suppressing HIF2α using an inhibitor (PT2567) or HIF2α knockout mice attenuated the cell proliferation processes, development of pulmonary hypertension, and RVH in rats exposed to 5 weeks of hypobaric hypoxia [66]. In contrast to these studies, we found that the exposure to long-term hypobaric hypoxia alone did not cause HIF2α overexpression. One of the possible explanations for this finding could have been a greater activation of calpain, which prevents the stabilization of HIF2α under intermittent and permanent hypoxia conditions [67]. On the other hand, increased levels of zinc inhibit calpain [68], which could respond to the effect of zinc administration in the (CIHz and CHz) conditions in this study. In addition, the controversies regarding HIF2α expression could be related to factors such as the time of exposure to hypoxia, type of murine model, tissue examined, and in vivo and in vitro model systems [69].

Zn is a factor that has been demonstrated to influence the development of cardiopulmonary pathologies under hypoxia exposure [11,21]. Our results showed that the exogenous administration of Zn produced a greater degree of RVH in chronic intermittent hypobaric hypoxia exposure, coincident with the further stabilization of HIF2α, which could also partly explain the RVH and HIF2α stabilization found in CH conditions. Therefore, our study suggested that HIF2α could induce the pathways that lead to cardiovascular damage when influenced by Zn under hypobaric hypoxia. This would be a novel result in CIH and CH exposure with Zn effects, considering that this is a particular model of exposure.

MTF-1 is a transcription factor considered to be a cytoplasmic Zn sensor that is, subsequently, translocated to the nucleus after metal induction and under stress conditions [70], such as that resulting from other metals, ROS, and hypoxia, where it has been shown that the activation of MTF-1 stimulates the expression of specific genes in the presence of low oxygen concentrations [31,32]. Regarding the Zn effects, it has been shown that Zn administration activates Zn transporters via the upregulation of HIF2α and MTF-1 [71], which could explain the HIF2α expression results in our research. Therefore, MTF-1 and HIF2α could act in synchronization under conditions of prolonged-term hypobaric hypoxia under the influence of Zn. However, in our study, there was also an increase in MTF-1, not only after exogenous zinc administration under hypoxic conditions (CIHz and CHz) but also in hypoxia without zinc participation (CIHs and CHs), which would indicate that MTF-1 is a direct sensor of hypoxia even more so than HIF, as described by other authors [72,73].

MT is transcriptionally activated by the essential metal Zn as well as by environmental stresses, such as hypoxia [31,32]. In our research, long-term exposure to hypobaric hypoxia without zinc did not result in MT overexpression (CIHs and CHs), with respect to (CH)z conditions, which could be concordant with other studies of long-term intermittent hypoxia that showed an inhibition of MT expression [74]. Our results can, therefore, be explained by the fact that the action of MT is strongly regulated in the first days (acute hypoxia) and decreases over time as the chronic inflammatory process progresses [75]. However, after the exogenous administration of Zn, we found that the overexpression of MT occurred. This is concordant with the results using an in vitro model system of renal cells, which showed that zinc induces MT, whereas hypoxia does not have much of an effect [76]. This finding in this research was expected, considering the great affinity that MT has for buffering, binding, and releasing Zn and thus its homeostatic regulation in the cell [77,78]. Additionally, we think that the effect of Zn is directly associated with the upregulation of the HIF2α/MTF-1/MT pathway. Therefore, we consider this to be a novel result, as it is the first time that this pathway has been described under long-term hypobaric hypoxic conditions and Zn.

ZIP12 expression was upregulated under chronic hypobaric hypoxia without the major effects of zinc administration. These results are consistent with those previously reported by Zhao et al. [11], who showed that the Zn transporter ZIP12 was induced in chronic hypoxia. They also demonstrated that the genetic disruption of the solute carrier family 39 member 12 (Slc39a12), which encoded ZIP12 and inhibited the elevation of intracellular Zn levels, while reducing pulmonary artery pressure, RVH, and vascular remodeling [11]. This result was expected, given that the role of Zn as a key inducer in the development of HPV has previously been shown [21]. Additionally, the development of pulmonary hypertension by monocrotaline induced ZIP12 overexpression, and silencing of this transporter reversed pulmonary artery remodeling [79]. These results supported the role of ZIP12 in the development of pulmonary hypertension due to hypoxia or another cause.

We expected to find a greater ZIP12 expression in (CH)z with zinc’s respect to the CHs group because it is known that the administration of zinc salts produces a fibrotic and proliferative effect in the lung [80]. However, the values remain similar. This could be explained by self-regulation due to the excess of zinc, such as the observed values of zinc in plasma in this research, but this hypothesis requires further investigation. It is possible that the level of oxidative damage in CH conditions is already very high, and the Zn pathway does not further contribute, as previously mentioned. 

Regarding the intermittent hypobaric hypoxia conditions, we observed a tendency to be upregulated, with or without Zn administration. Therefore, our results supported the role of ZIP12 overexpression in the upregulation of the HIF2α/MTF-1/MT/ZIP12 pathway in the development of RVH.

We also explored whether PKCε is involved in the development of RVH and proliferation since previous studies on hypoxia have reported that this protein is associated with the development of pulmonary vasoconstriction, cardiomyocyte enlargement and/or contractility events [21,22,24,55]. PKCε activation was mostly increased under long-term hypobaric hypoxia, except in the CIH group. Interestingly, we found that Zn administration increased PKCε activation in this group, which could explain the high degree of RVH found in the CIH group and corroborate the negative effects of Zn supplementation, which could be related to proliferation processes. In addition, in endothelial cells under acute hypoxia, the PKCε protein and enzyme activity increased, whereas the addition of a Zn chelator (TPEN) reduced the PKCε levels. Moreover, there is evidence that PKCε could be a therapeutic target, as its suppression may improve cardiovascular alterations due to chronic hypobaric hypoxia and the development of pulmonary hypertension [22]. All these results confirm that the observed PKCε activation under long-term hypobaric hypoxia and the greater increase by the influence of Zn could explain the upregulation of the HIF2α/MTF-1/MT/ZIP12/PKCε pathway that we had proposed to be involved in the development of RVH. Thus, these findings could generate new perspectives for the treatment of cardiopulmonary pathologies.

We also evaluated changes in the wall thickness and the number of PASMCs, as it is known as one of the principal sites of action for the development of HAPH, where PASMCs interact with several extracellular and intracellular molecules and participate in the mechanisms that lead to proliferation, characterized by an increase in the total number of cells [81]. In this study, we observed an increase in the wall thickness and number of PASMCs under hypobaric hypoxic conditions consistent with other models of hypobaric hypoxia [82]. It is important to highlight that we observed a greater increase in the CIH with zinc administration, coincident with the degree of RVH found. In addition, it is important to mention that the increased number of PASMCs observed showed the same tendencies in the expression of PKCε and HIF2α levels. These results could suggest a possible role of PKCε and HIF2α in the proliferation of PASMCs under long-term hypobaric hypoxia, especially in intermittent hypobaric hypoxia under zinc influence [83]. Although a significant increase in PASMCs was not observed in the chronic hypoxia group with zinc administration, the degree of RVH founded could be explained by a higher vasoconstriction product of exacerbated oxidative stress, as shown in the present study regarding lipid peroxidation, also previously demonstrated by us [57], and being highly implicated in hypoxia-associated pathologies [26]. This was likely due to the accumulation of ROS under hypoxic conditions, as demonstrated in cells and tissues [27,28]. Similar findings of oxidative damage biomarkers have been found in humans after acute or chronic exposure to high altitudes [29], as well as in studies in hearts from CIH and CH rats [26]. In normobaric hypoxia, it has been reported that Zn is required for the increase in ROS in cell culture and that Zn chelation inhibits excess ROS generation [84]. However, we did not find an effect from Zn administration in either NX rats or the hypoxic groups. It should be noted that there were marked differences between in vitro cell and in vivo animal studies under hypobaric hypoxia conditions. 

Thus, regarding the controversy on the antioxidant or pro-oxidant effects of Zn supplementation, our results indicate that Zn may not be adequate as a supplement since it may exacerbate the alterations in the cardiopulmonary circuit already induced by hypoxia under intermittent long term hypoxic conditions. It is important to mention that there are very few studies analyzing this aspect, and the present work provides new information on the effects of Zn under long-term hypobaric hypoxic conditions, with respect to oxidative damage to lipids. To corroborate the possible effects on VPH and RVH, it would be necessary to evaluate a larger number of biomarkers of oxidative status (NADPH oxidase, S-glutathionylation, and catalase, among others), as we previously proposed [25].

### Future Perspectives

This research opens the possibility to further study the influence of Zn at high altitudes over long periods in populations with hypobaric intermittent exposures. This is important to consider because there are Zn extraction mines, where miners are exposed to both Zn and intermittent hypoxia, in South America, especially in Peru. The negative influence of environmental Zn on cardiovascular disorders has been previously reported, such as chronic Monge’s disease (excessive erythrocytosis) [85]. Another consideration arising from the present study is the supplementation with Zn in athletes who seek to improve their athletic performance at high altitudes as a strategy to augment the EPO response [41], which normally has been evaluated in periods of acute hypoxia, whereas the effect of supplementation has not been assessed for long-term intermittent or chronic periods. Additionally, according to our data, Zn supplementation would not be suitable for workers under the conditions of intermittent exposure to hypoxia. 

## 4. Materials and Methods

### 4.1. Animal Model

In this study, 48 male adult Wistar rats (3 months old; body weight 251.6 ± 1.9 g) were obtained from the animal facility of the Institute of Health Studies of Arturo Prat University, Iquique, Chile. The rats were placed in individual cages at a temperature of 22 ± 2 °C and a circadian rhythm of 12 h of light and 12 h of darkness. Feeding consisted of 15 g/day (PMI LabDiet^®^, Prolab RMH 3000, Arden Hills, MN, USA) and water ad libitum. The rats were randomly distributed into three experimental groups as follows: normobaric normoxia (NX), which served as a sea level control (n = 16); chronic intermittent hypobaric hypoxia (CIH), with 2 days of exposure to hypobaric hypoxia, alternating with 2 days of exposure to NX (n = 16); and chronic hypobaric hypoxia (CH), which involved permanent exposure to hypoxia (n = 16). Each group was subdivided into two groups (n = 8) that received either a Zn sulfate solution (z) at 1% (5 mg/kg body weight) or a saline solution (s) intraperitoneally every 4 days, which was a low dose, considering that a 28–73 mg/kg intraperitoneal injection of Zn salts is the lethal dose (LD50) in rats [42,43].

Hypobaric hypoxia was simulated in a chamber at 428 Torr, which was equivalent to an altitude of 4600 m above sea level, where the rats were exposed for 30 days. Chamber conditions were as follows: internal flow of 3.14 L/min of air and humidity between 21 and 30%. The time of ascension from sea level to 428 Torr was 60 min. NX rats were located in the same room at sea level (760 Torr) and housed under the same chamber humidity conditions as the groups exposed to hypoxia. At the end of the exposure period (day 30), the rats were euthanized with an overdose of ketamine (0.9 mg/kg body weight), and the hearts and lungs were collected. The heart was first used for RVH measurements and then immediately fixed for histological examinations. The lungs were placed in liquid nitrogen and then stored at −80 °C for further analysis. These experiments were performed at Arturo Prat University (Chile). The animal protocol and experimental model were in accordance with Chilean law No. 20380 regarding animal experimentation and were approved by the Research Ethics Committee of Arturo Prat University, Iquique, Chile.

### 4.2. Body Weight and Hemoglobin

Body weight was assessed at day 0 and at day 30 immediately after removal from the chamber using an electronic scale (Acculab V-1200^®^ 127, Chicago, IL, USA). Likewise, in the same period, a 1-mL volume of blood was collected from the tail vein of rats under ketamine anesthesia (0.3 mg/kg body weight) after 10 h of fasting. Hemoglobin (Hb) was measured using a Coulter Electronics counter, Cell Dyn 3700, Abbot^®^, Santa Clara, CA, USA. Plasma was obtained by centrifugation at 2500 rpm for 30 min at 4 °C (5804 R Eppendorf AG R, Hamburg, Germany).

### 4.3. Zinc Analysis

Zinc levels were evaluated in plasma and in lung tissue with an atomic absorption spectrophotometer (FAAS (213.9 NM) Varian SPECTR AA 55B^®^, Australia Pty, Agilent Technologies, Santa Clara, CA, USA) at the Department’s laboratory of Chemical and Pharmaceutical Sciences of Arturo Prat University. Zn levels were determined in plasma samples at day 0 and day 30. After the exposure protocol, Zn levels in lung tissue were measured.

In plasma, a standard solution of zinc (titrisol Merck^®^ 1000 mg zinc) diluted with 5% glycerol (*v*/*v*) was made over a concentration range of 0.1 to 1.0 μg/mL zinc in acid medium (HUPP Suprapur). Plasma samples were acidified and diluted fivefold in nuclease-free water, and measurements were performed in duplicate.

To assess the concentration of Zn in the lung, samples were lyophilized, digested in acid, and filtered. A calibration curve was generated with standard zinc solutions (1000 mg/L zinc, Merck, Darmstadt, Germany) over the concentration range of 0.2 to 1.2 mg/mL Zn in acidic medium. Samples were acidified and diluted in nuclease-free water, and measurements were performed in duplicate.

### 4.4. Right Ventricular Hypertrophy and Cardiac Histology

RVH was evaluated using Fulton’s index (RV/LV + septum (g/g)), as described previously [86]. Then, the right ventricular tissue was fixed in 4% paraformaldehyde at room temperature overnight and cut transversely for histological analysis. Thereafter, tissues were dehydrated and embedded in paraffin. Paraffin-embedded tissue slices (5 μm thick) were stained with hematoxylin and eosin (H&E), images were captured by light microscopy, and the area was measured with the ImageJ software.

### 4.5. Western Blot Analysis

For protein analysis, 100 mg of lung tissue was homogenized (Stir-Pak R, Barrington, IL, USA) with 1 mL of RIPA lysis buffer containing a mixture of phosphatase and protease inhibitors: 4 mM PMSF, 10 µM leupeptin, 1 mM EDTA, 1 mM EGTA, 20 mM NaF, 20 mM HEPES, and 1 mM DTT. Then, the homogenates were centrifuged (5804 R Eppendorf AG R, Hamburg, Germany) at 12,000 rpm for 20 min at 4 °C, and the supernatant was extracted. For quantification of the total protein extracted, the Bradford reaction was used [87] with a BioPhotometer (Eppendorf AG R, Hamburg, Germany) at 590 nm, and samples were then stored at −80 °C. For Western blotting, the samples were previously diluted with Laemmli 2X (0.125 M Tris HCl, 4% SDS (*p*/*v*), 20% glycerol (*v*/*v*), 0.004% bromophenol blue, and 10% β-mercaptoethanol (pH 6.8)). The proteins were separated according to their molecular weight (MW) under an electric field via sodium dodecyl sulfate-polyacrylamide gel electrophoresis (SDS-PAGE) (30% bis-acrylamide (*v*/*v*), 150 mM Tris (pH 6.8 and 8.8), 1.0% TEMED (*w*/*v*), H_2_O). Electrophoretic separation was initiated with the application of direct current to 150 V over 90 min with a power supply (Polyscience R, EPS-300, Taipei, Taiwan, China), and the proteins were then transferred from the SDS-PAGE gel to a polyvinylidene fluoride (PVDF) membrane at 180 mA for 100 min with a semidry electroblotting system (OWLTM Separation systems, Panthe semidry Electroblotters, Thomas Scientific R, Barrington, IL, USA). To avoid nonspecific antibody binding, the membrane was blocked with bovine serum albumin (BSA) at a concentration range of 3–5% in TBS-T solution containing 10 mM HCl, 150 mM NaCl, and 0.05% Tween-20 at pH 7.4. The blocking time was 1 h at room temperature. Once the PVDF membrane was blocked, it was incubated with the corresponding primary antibody (EPAS1 (sc-28706), MT1a (sc-11377), MTF-1 (sc-48775), p-PKCε (sc-12355), PKCε (sc-214), ZIP12 (NBP2-75665) and β-actin (sc-130657)) at a dilution of 1:200 ((Santa Cruz Biotechnology R, Santa Cruz, CA, USA) and (Novus Biologicals, Littleton, CO, USA)) and incubated overnight at 4 °C. Finally, the membrane was incubated with secondary antibodies (anti-goat and anti-rabbit antibodies, Santa Cruz Biotechnology R, CA, USA) at a dilution of 1:2000 in 3% BSA for 1 h at room temperature and then washed with TBS-T and imaged in a dark room with a chemiluminescence kit (Chemiluminescence West Pico R, Supersignal Substrate, Thermo Scientific R, Rockford, IL, USA). Images were obtained with a luminescent image analyzer (ImageQuant LAS 500, Uppsala, Sweden), and the density of bands (peak areas) was analyzed and normalized according to β-actin expression, using ImageJ 1.48 v software (National Institutes of Health, Bethesda, MD, USA).

### 4.6. Confocal Microscopy

The total number of PASMC was determined by staining with the nuclear dye DAPI and confocal microscopy. After euthanasia, the removed lung was placed in a Petri dish and pulmonary artery branches (3rd order). The segments were incubated with the nuclear dye 4,6-diamidino-2-phenylindole (DAPI Molecular Probe D1306; 1:500 from 5 mg/mL stock; 30 min, RT in the darkness) and washed twice (30 min, RT). Then, the pulmonary arteries were cut in rings with a blade and mounted on a slide equipped with a small well made of spacers, filled with mounting medium (Citifluor, Aname, Spain), and covered with a cover glass.

The arterial rings were visualized with a Leica TCS SP2 confocal system (Leica Microsystems, Wetzlar, Germany) at the Universidad Autónoma de Madrid of Spain facilities, using the 488 nm/515 nm line. There, 1 μm thick serial images (25 μm in total) were captured with a 63× objective at zoom 2 in 3 randomly chosen areas of the ring at identical conditions of brightness, contrast, and laser power for all of the experimental groups. ImageJ image analysis software was used for quantification of total PASMC.

### 4.7. Lung Lipid Peroxidation

Lipid peroxidation in lung tissue was assessed through the determination of malondialdehyde (MDA) concentrations (µmol/L) using a colorimetric assay. First, 30 mg of lung tissue was homogenized in 400 µL of RIPA buffer (50 mM Tris-HCl, 1% Triton X-100, 150 mM NaCl, and 0.1% SDS) for 2 min at 4000 rpm with a homogenizer (Stir-Pak^®^, Brinton, IL, USA) at 4 °C. Then, 100 µL of the sample (homogenized tissue) was mixed with 200 µL of trichloroacetic acid (TCA; 10%) on ice for 30 min. Subsequently, the mixture was centrifuged at 4000 rpm for 15 min at 4 °C, and the supernatant (200 µL) was mixed with 200 µL of thiobarbituric acid (TBA; 0.67%) and incubated in a water bath (100 °C) for 1 h. Finally, the absorbance was measured with a spectrophotometer (Thermo Electron Corporation^®^, Madison, WI, USA) at 532 nm, and a calibration curve with MDA was used for quantification.

### 4.8. Data Analysis

All data recorded were included in a database and analyzed using the SPSS program (IBM^®^ SPSS R V.21.0R, Armonk, NY, USA). Mean and standard error (SE) values were calculated. Normality of the variables was established by the Kolmogorov–Smirnov test. As the variables had a normal distribution, parametric tests were used. To determine the change over time of the measured variables in each group, repeated measures analysis of variance was performed with a least significant difference (LSD) post hoc test. To establish the differences between groups, one-way ANOVA was performed, followed by an LSD post hoc test. The level of significance was established at the 95% confidence level (*p* < 0.05).

## 5. Conclusions

In this work, we demonstrated that both Zn administration and hypoxia might contribute to the development of RVH, most probably through exacerbation of the HIF2α/MTF1/MT/ZIP12/PKCε pathway, leading to HPV, under prolonged hypobaric hypoxia, including chronic intermittent exposure. Future studies will allow us to evaluate the role of this pathway in human populations exposed to these conditions to establish therapeutic targets for the mitigation of the exacerbated mechanism of HPV and the development of cardiovascular pathologies at high altitudes.

## Figures and Tables

**Figure 1 ijms-24-09567-f001:**
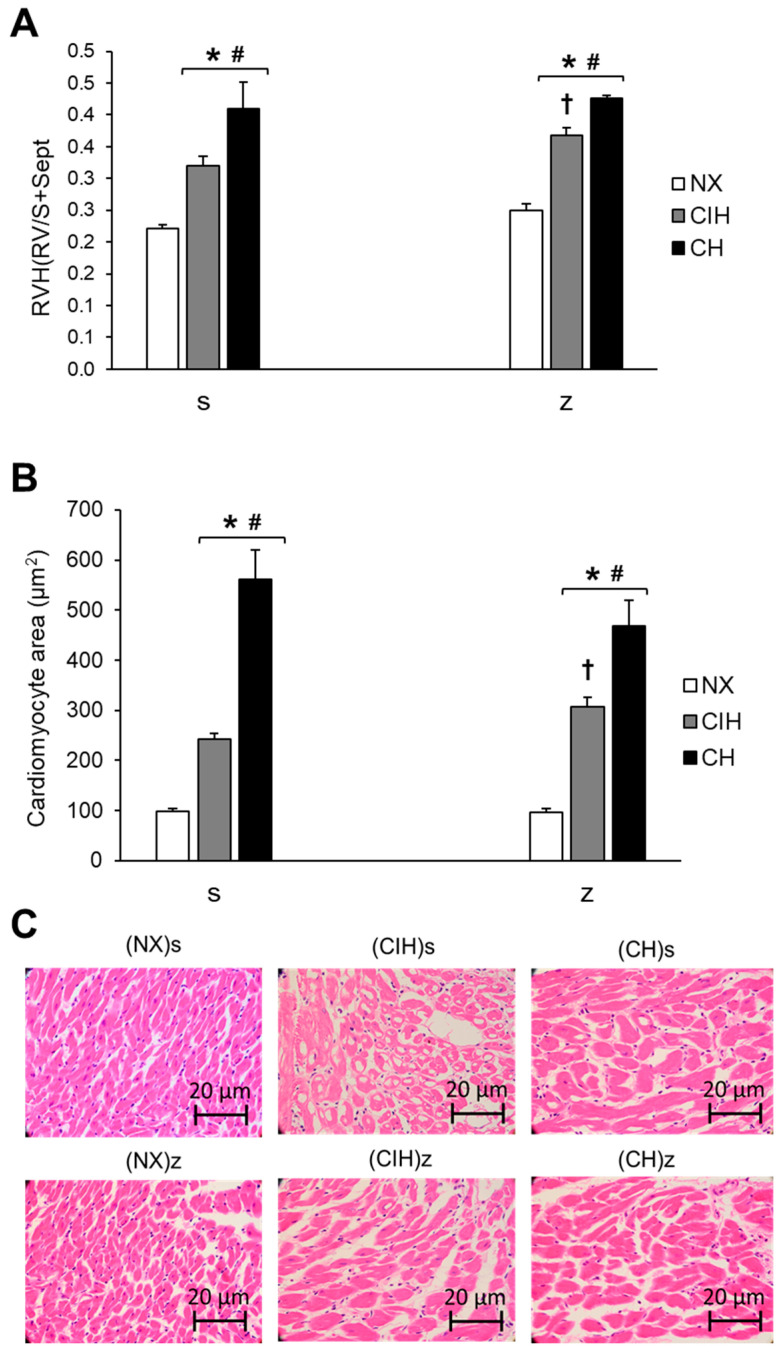
Measurements of RVH: (**A**) Fulton’s index (right ventricle (RV) weight/(left ventricle (LV) weight + septum weight)); (**B**) Area of cardiomyocytes (µm^2^); (**C**) Representative hematoxylin and eosin staining images of RV slices. Saline solution (s) and zinc (z) groups on day 30. Normoxic group (NX), chronic intermittent hypoxia group (CIH), and chronic hypoxia group (CH). The values are the mean ($\bar{X}$) ± standard error (SE). * *p* < 0.05 hypoxia vs. NX, # *p* < 0.05 CIH vs. CH, † *p* < 0.05 saline (s) vs. zinc (z).

**Figure 2 ijms-24-09567-f002:**
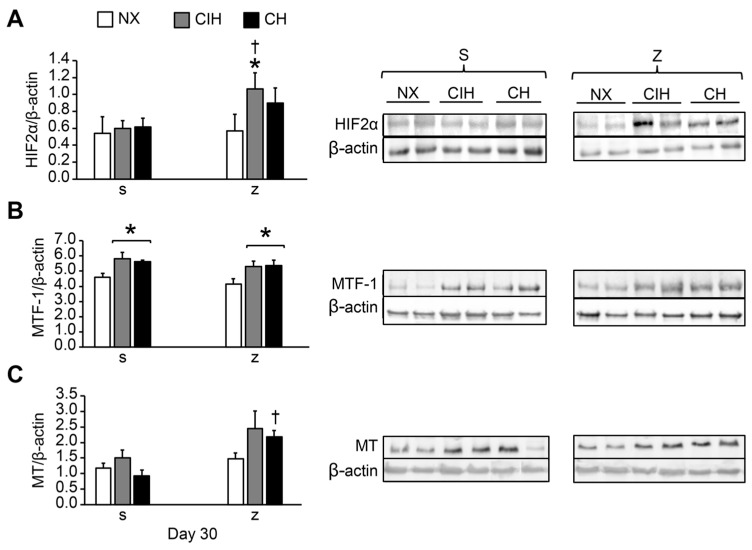
Protein expression levels in lung tissue: (**A**) HIF2α; (**B**) MTF-1; (**C**) MT. Saline solution (s) and zinc (z) groups on day 30. Normoxic group (NX), chronic intermittent hypoxia group (CIH), and chronic hypoxia group (CH). Representative bands are shown. The values are the mean ($\bar{X}$) ± standard error (SE). * *p* < 0.05 hypoxia vs. NX, † *p* < 0.05 saline (s) vs. zinc (z).

**Figure 3 ijms-24-09567-f003:**
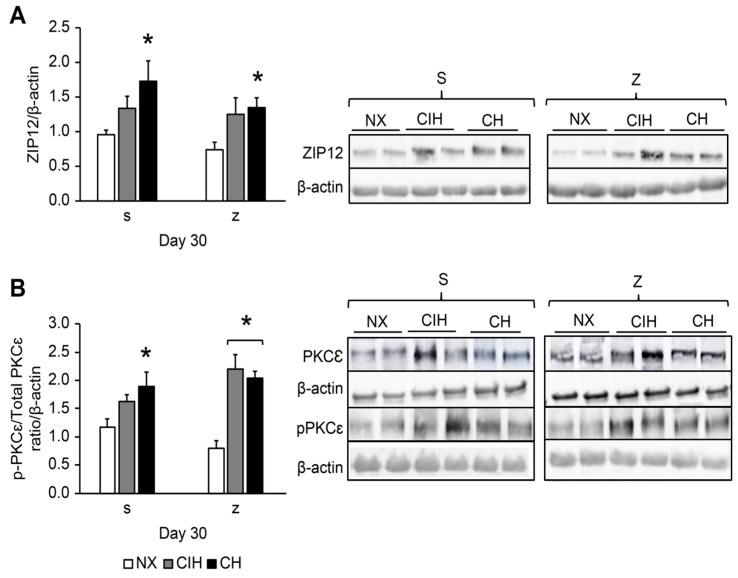
Protein expression levels in lung tissue: (**A**) ZIP12 protein; (**B**) PKCε activation (p-PKCε/total p-PKCε ratio). Saline solution (s)- and zinc (z)-treated groups on day 30. Normoxic group (NX), chronic intermittent hypoxia group (CIH), and chronic hypoxia group (CH). Representative bands are shown. The values are the mean ($\bar{X}$) ± standard error (SE). ***** *p* < 0.05 hypoxia vs. NX.

**Figure 4 ijms-24-09567-f004:**
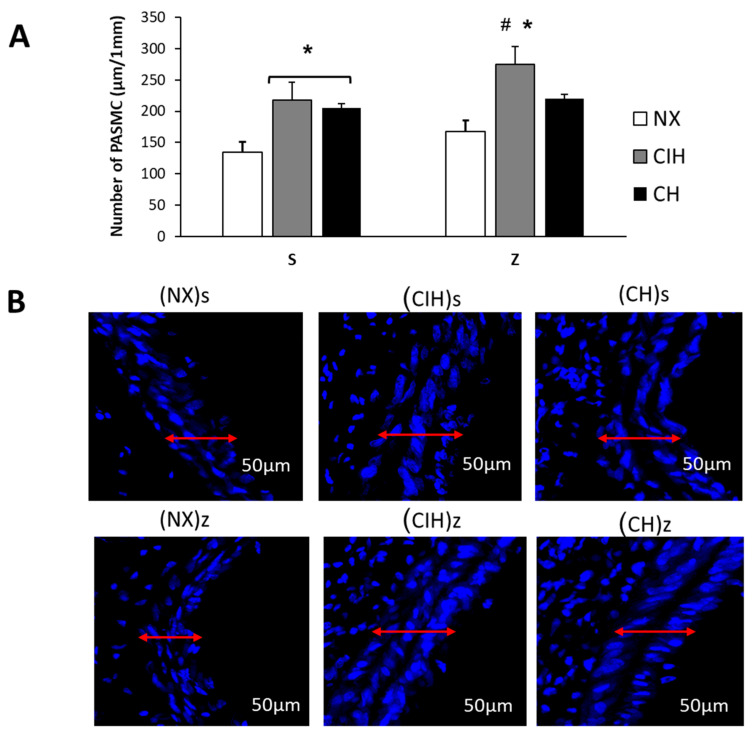
Number of pulmonary artery smooth muscle cells (PASMC). (**A**) Number of PASMC (µm/1 mm); (**B**) Representative confocal images, DAPI staining of number PASMC. Saline solution (s)- and zinc (z)-treated groups on day 30. Normoxic group (NX), chronic intermittent hypoxia group (CIH), and chronic hypoxia group (CH). Representative bands are shown. The values are the mean ($\bar{X}$) ± standard error (SE). * *p* < 0.05 hypoxia vs. NX, # *p* < 0.05 CIH vs. CH.

**Figure 5 ijms-24-09567-f005:**
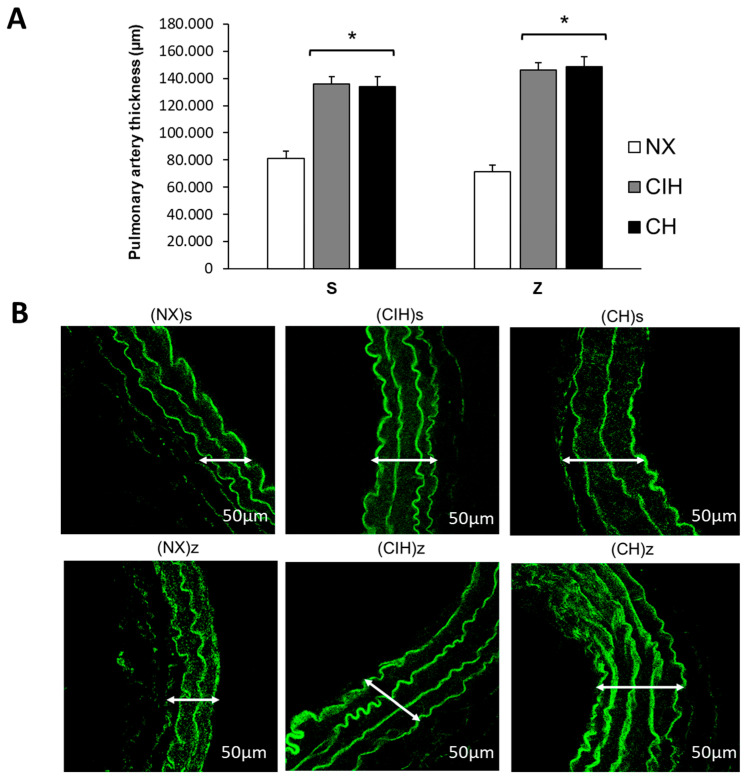
Pulmonary artery thickness. (**A**) Pulmonary artery wall thickness (µm); (**B**) Representative confocal images of the pulmonary artery wall thickness. Saline solution (s)- and zinc (z)-treated groups on day 30. Normoxic group (NX), chronic intermittent hypoxia group (CIH), and chronic hypoxia group (CH). Representative bands are shown. The values are the mean ($\bar{X}$) ± standard error (SE). * *p* < 0.05 hypoxia vs. NX.

**Figure 6 ijms-24-09567-f006:**
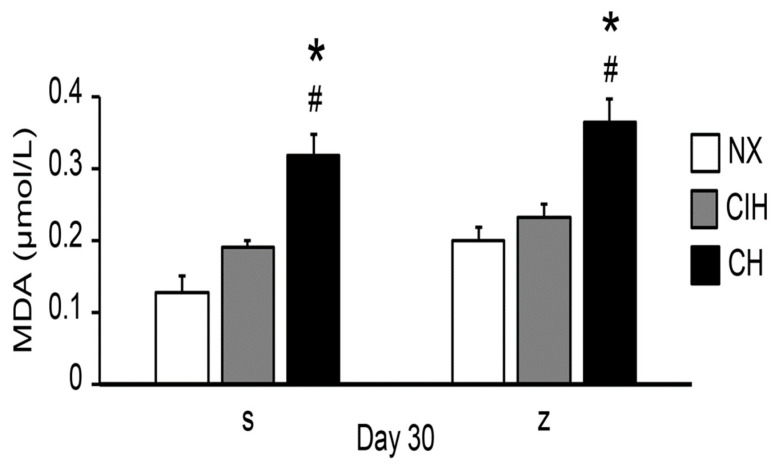
MDA levels in the saline solution (s) and zinc (z) groups on day 30. Normoxic group (NX), chronic intermittent hypoxia group (CIH), and chronic hypoxia group (CH). The values are the mean ($\bar{X}$) ± standard error (SE). * *p* < 0.05 hypoxia vs. NX, # *p* < 0.05 CIH vs. CH.

**Table 1 ijms-24-09567-t001:** Body weight, hemoglobin, and Zn levels in the plasma and lung at days 0 and 30.

VARIABLES	NX	CIH	CH
**BODY WEIGHT** **(gr)**			
Day 0 saline zinc	310.50 ± 10.31	303.13 ± 12.01	317.50 ± 7.21
	305.63 ± 9.46	322.38 ± 8.66	323.29 ± 11.67
Day 30 saline zinc	326.13 ± 7.95 &	277.38 ± 13.59 *&	295.38 ± 10.02 &
	328.00 ± 7.82 &	280.88 ± 13.72 *&	297.00 ± 11.60 &
**HEMOGLOBIN** **(mg/dL)**			
Day 0 saline zinc	14.92 ± 0.39	15.37 ± 0.47	14.83 ± 0.25
	15.44 ± 0.38	15.40 ± 0.24	14.42 ± 0.30 *#
Day 30 saline zinc	14.46 ± 0.32	21.05 ± 0.47 *&	23.30 ± 0.28 *#&
	14.91 ± 0.21	20.45 ± 0.40 *&	22.84 ± 0.52 *#&
**PLASMA ZINC** **(µg/mL)**			
Day 0 saline zinc	2.14 ± 0.16	2.27 ± 0.18	2.20 ± 0.17
	2.18 ± 0.19	2.12 ± 0.17	2.25 ± 0.15
Day 30 saline zinc	2.03 ± 0.21	0.90 ± 0.85 *&	0.74 ± 0.08 *&
	1.99 ± 0.09	1.44 ± 0.06 *&	1.25 ± 0.14 *&†
**ZINC LUNG** **(mg/mL)**			
Day 30 saline zinc	103.69 ± 3.95	93.83 ± 3.57	102.42 ± 3.76
	111.95 ± 3.44	91.66 ± 1.71 *	94.14 ± 2.24 *

Saline solution (s) and zinc (z) were administered to the normoxic (NX), chronic intermittent hypoxia (CIH), and chronic hypoxia (CH) groups. The values are presented as the mean ($\bar{X}$) ± standard error (SE). * *p* < 0.05 hypoxia vs. NX, # *p* < 0.05 CIH vs. CH, † *p* < 0.05 s vs. z, & *p* < 0.05 day 0 vs. day 30.

## Data Availability

Data availability http://medicinaenaltura.cl/, accessed on 23 May 2023.

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
