# Peer review of "Effects of Zinc on the Right Cardiovascular Circuit in Long-Term Hypobaric Hypoxia in Wistar Rats"

_ijms, 2023, doi:10.3390/ijms24119567_

Round 1

Reviewer 1 Report

The manuscript: Effects of Zinc on the Right Cardiovascular Circuit in Long- 2 Term Hypobaric Hypoxia in Wistar Rats studied an important problem regarding oxygen shortage in high mountain areas. The authors performed many experiments simulating chronic intermittent and chronic hypoxia. They supplemented with Zn different experimental groups but didn't explain the lack of significant results concerning CHS and CHZ groups data (RVH, Cariomyocytesn areas, MTF, MIF, ZIP, pKPKC,...). Besides, the authors in the discussion section didn't even discuss it. 

Minor concerns:

The authors should add the name and precise indications regarding the Image Analysis System they used.

The English should be improved.

Author Response

We appreciate your comments, and all your suggestions were amended and highlighted in yellow in the manuscript. "Please see the attachment".

Reviewer 2 Report

In this manuscript (ID#: ljms-2320476), entitled “Effects of Zinc on the Right Cardiovascular Circuit in Long-Term Hypobaric Hypoxia in Wistar Rats”, authors Arriaza et al studied the effect of zinc on the pathophysiology of hypobaric hypoxia using a rat animal model. Their results indicated that hypobaric hypoxia induced right ventricular hypertrophy and reduced plasma Zn levels. Treatment with Zn augmented those alterations and upregulated HIF/MTF/MT/ZIP/PKC pathway. The topic of study is interesting. However, the experimental design is not rigorous. The results are not reliable. Several major concerns are listed in the following paragraphs:

1. The zinc levels in plasma and lung were examined (table 1). It would be interesting that the zinc levels in the cardiac tissue. The hypoxia condition would also affect heart directly inducing cardiac hypertrophy.  

2. The lung tissue was used in this study. However, pulmonary tissue contains different cells, which perform their unique functions. Therefore, it would be better to isolate them to examine their alterations separately. For example, pulmonary arteries are very important in the development of pulmonary hypertension. At least, arterial hypertrophy and remodeling should be examined.

3. Fig 1 shows that zinc treatment significantly worsened CIH-associated cardiac hypertrophy. In this study, zinc treatment leaded to a damaging effect on pulmonary circulation.  It is well known that zinc supplement is able to suppress inflammation and against oxidative damages (reduce ROS levels) in so many disorders. Please address this controversy.

4. The protein expression in HIF/MTF/MT/ZIP/PKC pathway was studied. However, no enzymic activity was tested. In addition, their regulatory relationship was not evaluated. 

5. The data from this manuscript is not strong enough to support the conclusion shown in Fig 5. Those alterations in protein expression may be just associated with hypobaric hypoxia. Their role in this condition is still not clear. At least, the effect of enzyme inhibitor should be examined on hypoxia-induced HPV and RVH.

Author Response

We appreciate your comments and suggestions, which we have incorporated new data within what was possible perform and are highlighted in green in the manuscript. " Please see the attachment"

Round 2

Reviewer 1 Report

The authors have addressed my concerns.

Reviewer 2 Report

No additional recommendation